# Mean sea surface temperature changes influence ENSO-related precipitation changes in the mid-latitudes

Young-Min Yang [1], Jae-Heung Park [2], Soon-Il An [2,3 ✉], Bin Wang[1,4 ✉] & Xiao Luo[4]

El Niño profoundly impacts precipitation in high-population regions. This demands an advanced understanding of the changes in El Niño-induced precipitation under the future global warming scenario. However, thus far, consensus is lacking regarding future changes in mid-latitude precipitation influenced by El Niño. Here, by analyzing the Coupled Model Intercomparison Project simulations, we show that future precipitation changes are tightly linked to the response of each type of El Niño to the tropical Pacific mean sea surface temperature (SST) change. A La Niña-like mean SST change intensifies basin-wide El Niño events causing approximately 20% more precipitation over East Asia and North America via enhancing moisture transport. Meanwhile, an El Niño-like mean SST change generates more frequent eastern Pacific El Niño events, enhancing precipitation in North American. Our findings highlight the importance of the mean SST projection in selectively influencing the types of El Niño and their remote impact on precipitation.

[1] Earth System Modeling Center, Key Laboratory of Meteorological Disaster, Ministry of Education (KLME)/Joint International Research Laboratory of Climate and Environment Change (ILCEC)/ Collaborative Innovation Center on Forecast and Evaluation of Meteorological Disasters (CIC-FEMD), Nanjing University of Information Science and Technology, Nanjing, China. [2] Division of Environmental Science and Engineering, Pohang University of Science and Technology, Pohang, Korea. [3] Department of Atmospheric Sciences and Irreversible Climate Change Research Center, Yonsei University, Seoul, Korea. [4] Department of Atmospheric Sciences and International Pacific Research Center, University of Hawaii, Honolulu, HI, USA. ✉email: sian@yonsei.ac.kr; bwang@hawaii.edu

Considering the severe impact of the El Niño-Southern Oscillation (ENSO) on global[1,2] and regional weather[3–6], climate[2,3], and society[1,2], the improvement in the reliability of future projections of El Niño events and uncovering their underlying mechanism is of great importance to the climate science community[7–11]. Many studies have reported that ENSO's intensity and frequency might increase under greenhouse warming[9–13]. However, the intermodel spreads of the projected future change of ENSO are relatively large[10,12]. A few studies have shown an increased frequency of central Pacific (CP) El Niño events[8,11]. In contrast, other studies have shown a more frequent occurrence of eastern Pacific (EP) El Niño events[9,14] and stronger sea surface temperature (SST) variability in the eastern Pacific under greenhouse warming[9,12,13]. These results suggest the presence of considerable uncertainty in the prediction of future changes with regard to El Niño diversity[8,10,15].

ENSO-driven tropical precipitation anomalies are projected to increase significantly under the high emission scenario proposed by the Intergovernmental Panel on Climate Change[16–18]. Previous studies have discussed future changes in precipitation anomalies induced by projected ENSO in East Africa[16,19–22], the Maritime continent[23–25], South and East Asia[17,19,26], North America[18–20], and major monsoon regions[17,19] based on enhanced ENSO-teleconnection under anthropogenic forcings. However, a few studies have suggested that the signal for consistent strengthening was relatively weak across the models[17,20,22], although the multi-model ensemble mean showed robust increases in ENSO-induced precipitation. A recent study[17] found that in the Coupled Model Intercomparison Project (CMIP) version 6, an El Niño-like eastern Pacific warming reduced North American monsoon rainfall as a result of the equatorward shift of the intertropical convergence zone. However, owing to large uncertainties in estimating the changes in SST, future changes in El Niño-induced precipitation in mid-latitude land regions have not been well addressed.

Recent studies[12,27] have shown that the El Niño behavior in the climate models that predict La Niña-like mean SST change differs from that in the models predicting an El Niño-like mean-state change[27], mainly because of the differences in the zonal SST gradients and upper-ocean stratification. Historically, observed warming in the western Pacific may induce more frequent and extreme El Niño events with warm anomalies over the central and eastern Pacific[12]. These results signify the importance of mean SST change for the accurate prediction of future changes in El Niño-induced precipitation anomalies.

Here, we show that the future changes in the mid-latitude precipitation are closely related to how different types of El Niño and their teleconnection will change in future. Furthermore, analysis of the CMIP5 model simulations reveals that the changes in different types of El Niño are determined by future changes in tropical Pacific mean SST. In La Niña-like mean-state SST change, the basin-wide type of El Niño event is intensified via the enhanced zonal advection feedback; such an event induces anticyclonic flow in the western Pacific and the tropical Atlantic Ocean through atmospheric teleconnections. This anticyclonic flow enhances the northward moisture transport to East Asia and North America, contributing to an increase in precipitation by approximately 20%. On the other hand, an El Niño-like mean-state SST change generates more frequent EP-type El Niño events, inducing more precipitation over North America (an increase of up to 10–15%).

## Results

### Two projections of mean SST gradient change in the Pacific.

Figure 1 shows the relationship between tropical mean SST change and zonal SST gradient changes (zonal SST gradient is calculated as the difference of mean-state SST in the western Pacific [155°E–175°W] and the mean-state SST in the eastern Pacific [115°W–145°W]; see "Methods" section) obtained from observations and historical and future scenario runs of CMIP5 models. The observed change (i.e., the difference between a recent period [1980–2005] and a past period [1940–1970]) shows moderate warming in the western Pacific compared to the eastern Pacific (a La Niña-like mean SST change). In historical simulations, the models with La Niña-like and El Niño-like mean SST changes account for approximately 40% each of the total models used in this study, whereas the models with a relatively small (or neutral) SST change account for approximately 20% (Fig. 1a). Under global warming, the zonal SST gradient changes (differences between historical and RCP 4.5 [or RCP8.5] runs) derived

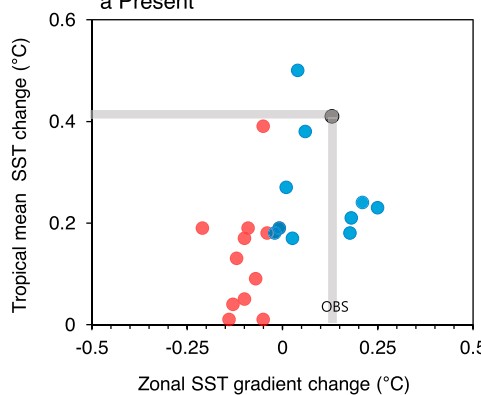
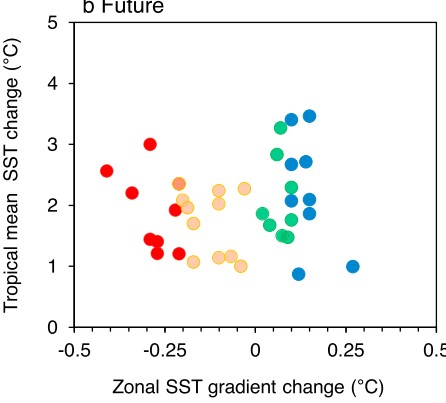

**Fig. 1 Two projections of zonal SST gradient change. a** Relationship between mean sea surface temperature (SST) changes in the tropical Pacific (y-axis, unit: °C) and zonal SST gradient changes (x-axis, unit: °C) in climate models during present periods. **b** Same as **a** but for future periods. For present periods, the differences between the mean SST (or zonal SST gradient) of 1940–1970 and 1980–2005 are used. The zonal SST gradient is measured as the difference between the western Pacific SST (5°S–5°N, 155°E–175°W) and the eastern Pacific SST (5°S–5°N, 115°W–145°W). The Extended Reconstructed Sea Surface Temperature version 5 dataset is used for observation. The observations are marked with a gray line. For future periods, the differences between tropical mean SST (20°S–20°N) of the Representative Concentration Pathway (RCP) 4.5/8.5 (2038–2087) scenario and that of the historical simulation (1943–1992) are used. The zonal SST gradient changes are defined as the differences of zonal SST gradient between RCP4.5/8.5 and historical scenarios. Red (orange) dots represent the models experiencing strong (weak) El Niño-like mean SST changes, while blue (green) dots indicate strong (weak) La Niña-like mean SST changes in the future.

from CMIP5 models are largely scattered (−0.42 to 0.29 °C). Notably, the zonal SST gradient change tends to be independent of the tropical mean SST change. In this study, we classified climate models into two groups based on the zonal SST gradient changes: 1) positive zonal SST gradient change (La Niña-like mean SST change; LN-Models) and 2) negative zonal SST gradient change (El Niño-like mean SST change; EN-Models). Under future global warming, the LN-Model (EN-Model) accounts for approximately 40% (50%) of the total models used in this study. The model with a relatively small (or neutral) SST change is approximately 10%, suggesting that the future change in the zonal SST gradient may be uncertain (Fig. 1b).

**El Niño diversity associated with El Niño-like mean SST change.** To examine the change in El Niño diversity and explore the relationship between zonal SST gradient change and each El Niño event, we applied cluster analysis to historical and high emission scenario simulation data from 18 climate models (see "Methods" section). Based on the cluster analysis, El Niño events were classified into the following four types: 1) basin-wide (BW), 2) EP, 3) CP, and 4) successive. We concentrate on the BW, EP, and CP types of El Niño events in this study.

The evolutionary composites of three distinct El Niño events from the observation and the models are shown in Fig. 2. The observation shows that, in the EP El Niño events, cold anomalies occur during the previous winter and spring (Fig. 2b). They drastically change to warm anomalies from the far eastern Pacific during early summer and then propagate westward and reach the maximum in December around 130 °W. These changes increase the thermocline depth in the eastern Pacific, inducing rapid warming from the eastern to central Pacific. Compared to the observations, the EN-Models' historical simulations capture the observed evolution of equatorial SST (Fig. 2e): cold anomalies in the spring and warm anomalies from summer to winter, and westward propagation of warm SST (Supplementary Fig. 3a). However, the maximum of the warm anomalies is relatively weak and shifts westward. When anthropogenic forcing was exerted in the EN-Models, EP El Niño events tended to intensify (Fig. 2h). The stronger anomalous westerlies in the eastern Pacific (Supplementary Fig. 2a) may be related to the greater thermocline depth (Supplementary Fig. 3a).

We observed BW El Niño events starting in the western Pacific during the previous winter and propagating eastward with a rapid basin-wide extension (Fig. 2a). BW El Niño shows a very large maximum amplitude in the eastern Pacific; it may be developed by both zonal advective feedback in the central Pacific and thermocline feedback in the eastern Pacific. To examine the dominant dynamic processes of each type of El Niño, we conducted an ocean mixed-layer heat budget analysis (see "Methods" section). The zonal advective and thermocline feedbacks were dominant in the BW El Niño event (Supplementary Table 1).

The EN-Models reasonably reproduce the observed pattern of SST anomalies: the initiation of warm anomalies in the western Pacific, eastward propagation, and rapidly increased warm SST anomalies (Fig. 2d). However, the models slightly overestimate the maximum SST anomalies and produce a long-lived peak in the winter season. The westerly anomalies occur in the initial phase (e.g., westerly wind burst events), and both SST warming and a deep thermocline are generated, which move into the central Pacific. Under the global warming scenario, the warm SST anomalies are reduced significantly and become meridionally narrow in the equatorial eastern Pacific (Fig. 2g). The El Niño-like mean SST change induces weakened anomalous westerlies in the western and central Pacific by weakening the associated

convective anomalies (Supplementary Fig. 2a). These changes may reduce zonal advective feedback in the western Pacific and thermocline feedback in the eastern Pacific, resulting in a decrease in the intensity of the BW type of El Niño.

For CP El Niño, the observation shows similar temporal evolution to that of BW El Niño; however, its magnitude are relatively weak (Fig. 2c). The EN-Models capture the initiation of warm SST anomalies in the western Pacific but fail to simulate eastward propagation; the model shows both eastward propagation from the western Pacific and westward propagation in the eastern Pacific (Fig. 2f). The changes in SST during CP El Niño caused by global warming are also similar to those of BW El Niño. The magnitude of the CP El Niño decreases with increasing anthropogenic forcings (Fig. 2i), indicating that the zonal SST gradient changes affect this type of El Niño.

**El Niño diversity associated with La Niña-like mean SST change.** In the LN-Models, the temporal evolutions of the BW and CP El Niño events from the historical runs resemble those of the EN-Models, although their magnitudes during the mature phase are relatively weak (Fig. 2j, i). With anthropogenic forcings, the warm SST anomalies of the BW (or CP) types of El Niño are intensified and broadened outward from the equator (Supplementary Fig. 1b). The cold anomalies in the western Pacific and the eastern Indian Ocean are slightly reduced. Stronger anomalous westerlies in the western Pacific may contribute to more intensive BW (or CP) El Niño (Fig. 2m, 2o). These results suggest that a positive change in the zonal SST gradient may lead to strong BW (or CP) El Niño events via enhanced zonal advective feedback (Supplementary Table 1).

The evolutionary structure of EP El Niño events in the LN-Models is quite similar to that in the EN-Models (Fig. 2k). In the global warming simulation, the magnitudes of SST anomalies are weakened and the westward extension of warm anomalies is reduced (Fig. 2n). The reduced mean westerly surface wind in the eastern Pacific may contribute to weak EP El Niño by reducing the thermocline and enhancing upwelling feedback.

We hypothesized that the zonal SST gradient change may be a critical factor in determining the future changes in the frequency and intensity of the three types of El Niño events; the hypothesis was examined using 18 climate models. As shown in Fig. 3, when the zonal SST gradient increases under anthropogenic forcing, both the intensity and frequency of BW El Niño events significantly increase with a correlation coefficient of 0.76 and 0.70, respectively. Similarly, the intensity and frequency of CP El Niño increase in proportion to the zonal SST gradient. This implies that when anthropogenic forcings will induce La Niña-like warming in the equatorial Pacific, BW (and/or CP) El Niño events will intensify and occur more frequently. On the other hand, when the zonal SST gradient will decrease in the future, the frequency and intensity of the EP-type El Niño will increase with a correlation coefficient of −0.74 and −0.61, respectively.

**Precipitation changes under La Niña-like mean SST change.** The zonal SST gradient change may affect the projected precipitation by changing the characteristics of El Niño diversity. We compare the impacts of BW and EP El Niño events in Fig. 4 as the changes in the intensity of these types of El Niño events by anthropogenic forcings are relatively strong. Note that we applied a high-pass spectral filter (13 years) for eliminating the global warming trend and long-term climate variability and to focus on the interannual ENSO influence on precipitation.

Figure 4a shows a composite of precipitation anomalies from the observations during BW El Niño events. The observations show

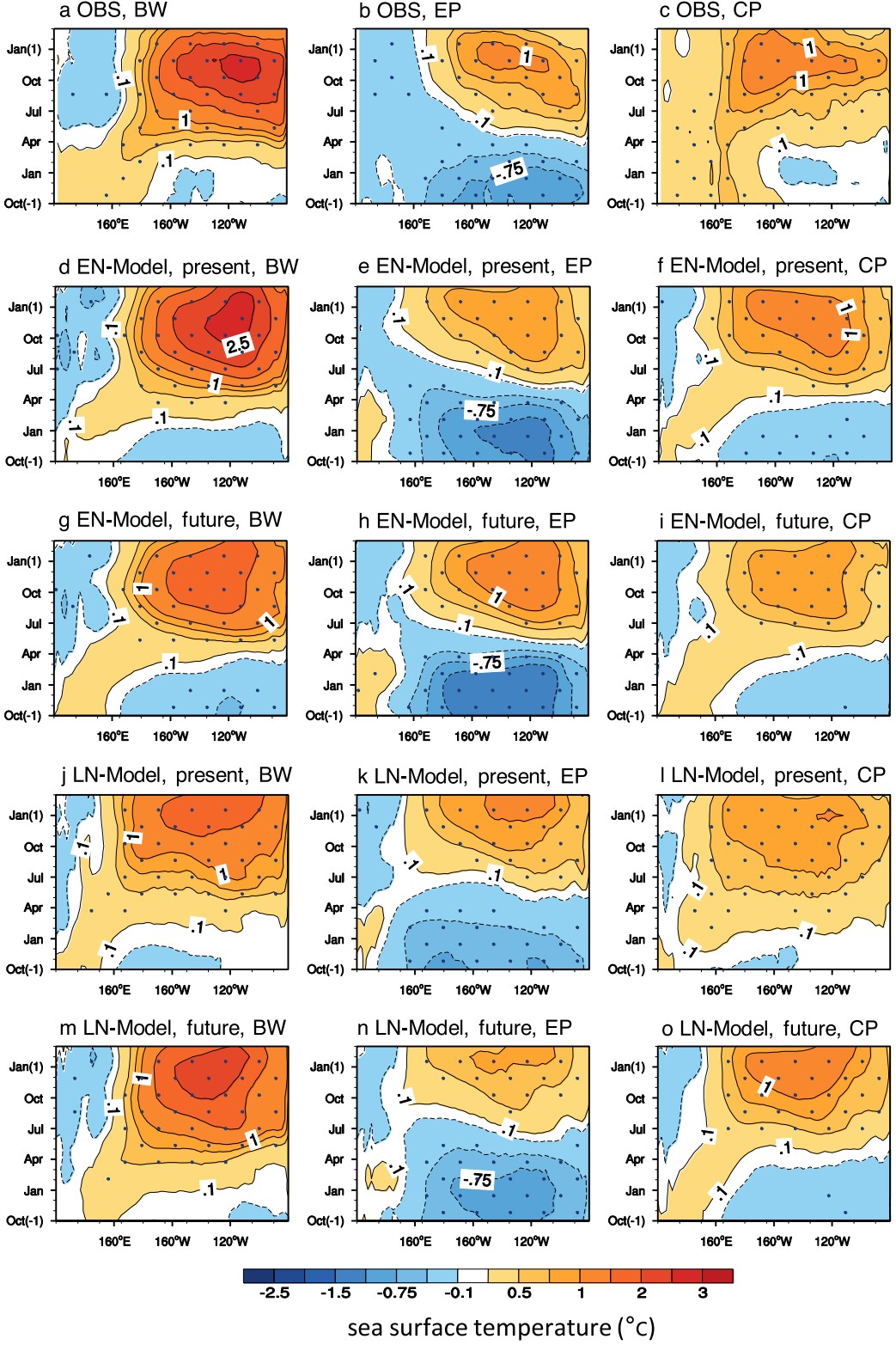

sea surface temperature (°C)

contour: sea surface temperature (°C)

strong, dry anomalies in Australia, the Amazon, and South Africa, and wet anomalies in the central and eastern U.S., East Asia, and East Africa. The LN-Models can capture observed historical precipitation anomalies over both tropical and mid-latitude land areas (Fig. 4c). In tropical oceans, strong, wet anomalies are observed in the central and eastern Pacific, whereas dry anomalies

are observed in the western Pacific, eastern Indian Ocean, and the tropical Atlantic Ocean. In the tropics, the horizontal patterns of precipitation anomalies are similar to those of the corresponding SST anomalies (e.g., Supplementary Fig. 1a). Regarding tropical land, wet (dry) anomalies occur in West Africa (the Amazon). The warm SST anomalies associated with BW El Niño generate rising

**Fig. 2 El Niño diversity associated with mean SST change. a–c** Longitude-time evolution of the equatorial Pacific sea surface temperature (SST) anomalies (shading and contour, units: °C) under observation for three Basin-wide (BW), Eastern Pacific (EP), and Central Pacific (CP) types of El Niño evolution. Each panel illustrates the composite of the equatorial SST anomalies averaged between 5°S and 5°. The time axis starts from October of the year before the El Niño year (−1) to February after the El Niño year (1). **d–f** Same as **a–c** but for present periods of the climate models with El Niño-like mean SST change (EN-Model). The present represents the historical run (1943–1992) from 18 climate models with two ensemble members. **g–i** Same as **d–f** but for the future periods, which shows a composite of both Representative Concentration Pathway (RCP) 4.5 (2038-2087) and RCP8.5 (2038–2087). **j–l** Same as **d–f** but for present periods of the models with La Niña-like mean SST change (LN-Model). **m–o** Same as **j–l** but for the future periods. All data are applied to a high-pass spectral filter (13 years) to eliminate global warming and long-term climate variability. The stippling denotes the regions where the signal (group mean) is larger than the noise (one standard deviation from the group mean of each member).

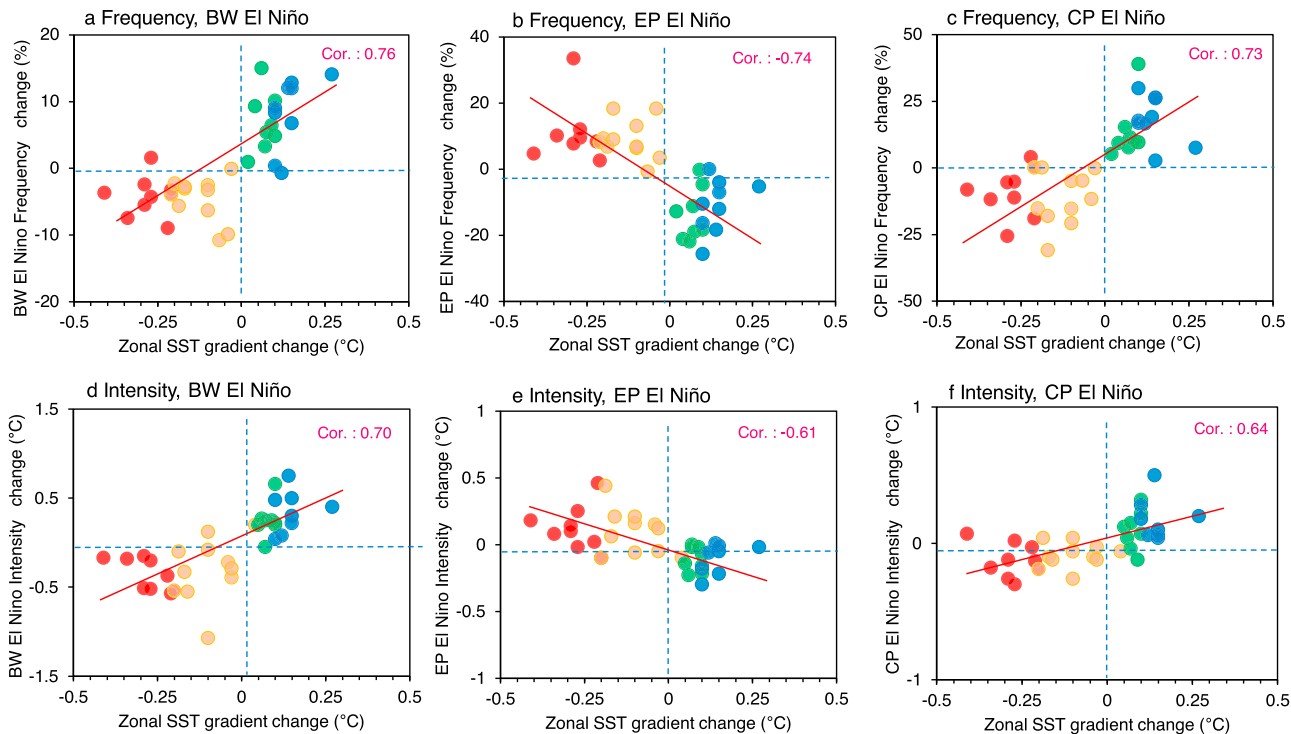

**Fig. 3 Relationship between zonal SST gradient and frequency/intensity change according to El Niño type. a–c** Relationship between zonal sea surface temperature (SST) gradient changes over equatorial Pacific (*x*-axis, units: °C) and frequency change of the Basin-wide (BW), Eastern Pacific (EP), and Central Pacific (CP) types of El Niño, respectively (*y*-axis, units: %). The zonal SST gradient is defined as the SST difference between the western Pacific (5°S–5°N, 155°E–175°W) and the eastern Pacific SST (5°S–5°N, 115°W–145°W). **d–f** Same as **a–c** but for the intensity change of the three El Niño types (*y*-axis, units: °C). The intensity is measured by boreal winter (October to February) SST anomalies averaged over 5°S–5°N and 80°W–180°W. The frequency (or intensity) changes are differences between historical (1943–1992) and Representative Concentration Pathway (RCP) 4.5/8.5 (2038–2087) scenarios. Red (blue) dots represent the models with El Niño-like (La Niña-like) mean SST change. Eighteen climate models were used. The linear regression line is shown as the solid red line in each panel, with correlation coefficient. Red (orange) dots represent the models experiencing strong (weak) El Niño-like mean SST changes, while blue (green) dots indicate strong (weak) La Niña-like mean SST changes in the future.

motions, with strong upper-level divergent flows being connected to upper-level convergences in the western Pacific and eastern Indian Ocean and the tropical Atlantic Ocean, where sinking (descending) motions are dominant (Supplementary Fig. 4a). These descending motions may contribute to suppressing convection resulting in less precipitation. The descending motions in the western Pacific produce surface easterly anomalies over the Indian Ocean, whereby the transport of moisture to Central Africa is enhanced and may lead to wet precipitation anomalies. The anomalous descending branch of the Walker circulation over the equatorial western Pacific gradually builds up with the aid of colder SST anomalies and low-level divergence, reinforcing warming in the central and eastern Pacific. The mid-latitudes of East Asia, North America, and Eastern Europe are characterized by wet anomalies. The descending motions in the western Pacific and the

tropical Atlantic Ocean generate anticyclonic wind anomalies and induce northward transport of moisture from the tropics to mid-latitude regions, which may contribute to more precipitation over East Asia and the eastern part of North America.

Under anthropogenic forcing, the wet anomalies in the central and eastern Pacific increase due to the warmer SST anomalies during an El Niño event (Fig. 4e). Notably, we hypothesized that the mean-state SST change due to anthropogenic forcing may induce greater El Niño intensity and frequency, leading to changes in precipitation. Stronger wet anomalies in the central and eastern Pacific increase the ascending motions and enhance the descending motion in the western Pacific and Amazon regions, inducing dry anomalies. On the other hand, the enhanced descending motion in the western Pacific and tropical Atlantic Ocean generates stronger anticyclonic anomalies, which increases the

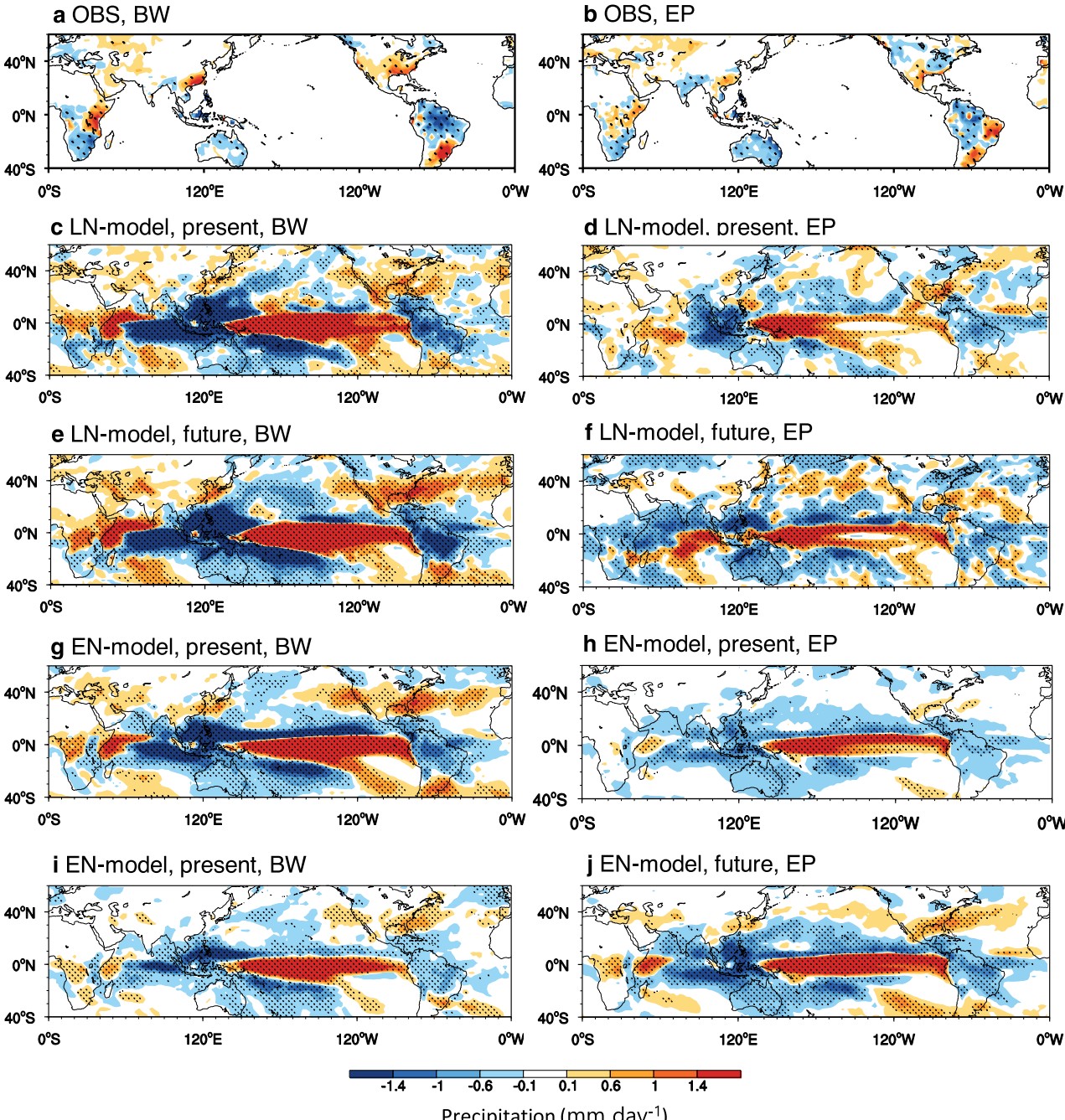

**Fig. 4 Future precipitation changes associated with El Niño diversity. a**, **b** Composites of boreal winter (October to February) precipitation anomalies (shading, mm day$^{-1}$) induced by Basin-Wide (BW) and Eastern Pacific (EP) types of El Niño derived from observations. **c**, **d** Same as **a**, **b** but for the present run of the model with La Niña-like mean SST change (LN-Model). The present represents the historical run (1943–1992) from the 18 climate models with two ensemble members. **e**, **f** Same as **c**, **d** but for the future period from both Representative Concentration Pathway (RCP) 4.5 and RCP8.5 (2038-2087). **g**, **h** Same as **c**, **d** but for the present run of the model with El Niño-like mean SST change (EN-Model). **i**, **j** Same as **g**, **h** but for the future period. All data are applied to a high-pass spectral filter (13 years) to eliminate global warming and long-term climate variability. The stippling denotes the regions where the signal (group mean) is larger than the noise (one standard deviation from the group mean of each member).

northward transport of moisture. The strong cyclonic flows in the northeastern Pacific generate northeastward wind in the eastern Pacific. Both enhanced northward circulations may contribute to more precipitation in East Asia and North America.

**Precipitation changes under El Niño-like mean SST change.**
The BW El Niño-induced precipitation anomaly pattern from the EN-Models resembles that from the LN-Models. The wet

anomalies in the central and eastern Pacific and dry anomalies in the western Pacific, eastern Indian Ocean, and the tropical Atlantic Ocean are observed from the simulations; although their magnitudes are relatively stronger (Fig. 4g). However, precipitation changes caused by anthropogenic forcings show almost contrasting trends (Fig. 4i). Under anthropogenic forcing, wet anomalies in the Pacific are reduced because of the correspondingly weakened SST anomalies. The rising motion induced by less warm SST is weakened, which induces a weak sinking motion,

resulting in fewer dry anomalies in the Indian, tropical Atlantic, and Amazon regions (Supplementary Fig. 4b). Weak anticyclonic anomalies occur due to reduced descending motion in the western Pacific and the tropical Atlantic Ocean, which induce weak northward transport of moisture, resulting in less precipitation in East Asia and North America. These results suggest that the zonal SST gradient change may be critical for assessing the projected precipitation change in mid-latitude regions, because it controls the magnitude of the BW El Niño event.

The impact of the zonal SST gradient change on precipitation during EP El Niño is shown in Fig. 4b. Observations reveal dry anomalies in Australia, the Amazon, and South Africa, and wet anomalies in the central U.S., East Asia, and East Africa. The LN-Models produce the observed horizontal pattern of precipitation anomalies (Fig. 4d). The wet precipitation anomalies occur in the western Pacific and extend along the equator. Dry anomalies occur in the Indian Ocean and the tropical Atlantic Ocean via a descending motion (Supplementary Fig. 4b). Both strong cyclonic flows in the northern Pacific and the southeastward wind in the far eastern Pacific cause northward moisture transport from the tropics to North America, which may increase precipitation in this region. In contrast, in South China, dry anomalies occur during the EP El Niño events. Under anthropogenic forcing, anomalous cyclonic flows in the northern Pacific are weakened and shift northward due to less warm SST anomalies in the equatorial Pacific (Fig. 4f). This reduces the northeastward wind in the eastern Pacific, resulting in less precipitation in North America. However, precipitation in East Asia increases because of anthropogenic forcing, which may be due to strong moisture transport by the enhanced anomalous southwestly wind from the subtropical western Pacific and westerlies from Saudi Arabia to East Asia through India.

The horizontal pattern of EP El Niño-induced precipitation from the historical simulation of the EN-Models is similar to that of the LN-Models; however, both wet and dry anomalies are weaker (Fig. 4h). The wet anomalies in East Asia and North America are very weak, and might be attributed to the weakened teleconnection owing to reduced tropical precipitation. Compared to the observations, the climate models reproduce EP El Niño-induced precipitation anomalies in tropical land areas (e.g., the Amazon, Australia, and Central Africa) well, but have certain limitations in mid-latitude land areas. Under global warming forcing, the wet anomalies and ascending motions increase due to warmer SST anomalies, generating strong dry anomalies in the Indian Ocean and tropical Atlantic region via enhanced descending motions (Supplementary Fig. 4b). The increased northward transport of moisture from both enhanced anticyclonic winds in the North Pacific and anticyclonic winds in the tropical Atlantic Ocean may contribute to more precipitation in North America. More precipitation anomalies in East Asia may enhance northward moisture transport by anticyclonic winds in the western Pacific (Fig. 4j).

To distinguish between the El Niño-induced and mean SST-induced changes in precipitation, we calculated the precipitation anomalies during normal (or neutral) years using the models (Supplementary Fig. 5). The results show no significant changes in precipitation in mid-latitudes between historical and future climate simulations, suggesting that the changes in precipitation (shown in Fig. 4) are primarily due to the El Niño change projected by global warming.

Based on the projected change in precipitation classified by each type of El Niño event and the zonal SST gradient change, we hypothesized that precipitation changes in North America and East Asia during BW and EP El Niño events might be dependent on the changes in the zonal SST gradient. We tested this hypothesis using

the historical runs of the 18 models and future precipitation projections (Fig. 5). When the zonal SST gradient increased under anthropogenic forcing, the intensity of precipitation due to El Niño events increased significantly in East Asia and North America with a correlation coefficient of 0.74 and 0.71, respectively. This implies that if anthropogenic forcing enhances the SST gradients in the central Pacific, more precipitation anomalies will occur over East Asia and North America by intensified and frequent BW El Niño. In contrast, during EP El Niño events, precipitation in North America is reduced with an increase in the zonal SST gradient with a correlation coefficient of −0.62. Notably, there are less significant precipitation changes in East Asia (correlation coefficient: −0.38).

## Discussion

This study demonstrated a strong relationship between the zonal SST gradient change and future changes in El Niño-induced precipitation. With an increase in the zonal SST gradient (as observed for the 1980–2019) extreme El Niño will occur more frequently, which in turn, will significantly enhance precipitation than that observed for the historical simulation (up to 20%) in East Asia and North America. On the other hand, many climate models simulate reduced zonal SST gradient changes, which would generate more frequent EP El Niño events and stronger precipitation in North America (10–15%). These results suggest that increasing anthropogenic forcing may induce greater El Niño-induced precipitation in the high-population mid-latitude regions.

We also investigated the relationship between the tropical mean SST change and El Niño intensity change; it was observed that the former does not contribute to a change in the intensity of the three types of El Niño events by global warming projections. It is also important to perform a reexamination using the CMIP6 model because the ENSO performance may be improved.

## Methods

**Observed data**. For the monthly mean SST, we used the National Oceanic and Atmospheric Administration Extended Reconstructed SST version 5[28]. Ocean temperature and thermocline depth data derived from the European Center for Medium-Range Weather Forecasts ocean reanalysis and ocean heat content data-sets were used as observation data[29]. Wind and precipitation data were extracted from the National Centers for Environmental Prediction dataset from NOAA-CIRES Twentieth Century Reanalysis v3 (1943–1992)[30].

**Definition of El Niño years**. We used 18 models with two ensemble numbers. To eliminate global warming and long-term climate variability and to focus on the interannual ENSO effect, we applied a high-pass spectral filter (13 years) to the CMIP5 dataset[23]. The first and last 13 years from the total period were discarded to remove edge effects caused by the spectral filter. Therefore, only 50 years of historical (1943–1992) and the representative concentration pathway (RCP) 4.5 (or RCP 8.5) scenarios (2038–2087) were used for the analysis. To increase the number of different types of El Niño events, two ensemble members were utilized (100-year data were used for model analysis). The SST anomaly averaged in the Niño 3.4 region (5°N–5°S, 120°W–170°W), known as the Oceanic Niño Index (ONI), was averaged for October–February (ONDJF) to identify the El Niño years. An El Niño year is defined if the ONDJF ONI is greater than or equal to 0.6 °C[12]. The same methods were applied to observations.

**Cluster analysis and ocean mixed-layer heat budget analysis**. Cluster analysis considers the temporal evolution of ENSO events from the onset to the mature phase, which is depicted by SST anomalies averaged between 5°S and 5°N. The K-means cluster analysis focuses on different space-time structures of the El Niño event SST anomalies. In the K-means cluster analysis, the squared Euclidean distance was used to measure the similarity between each cluster member and the corresponding cluster pattern. The silhouette clustering evaluation criterion was used to evaluate the performance of the cluster analysis. A high silhouette value indicates that the member is well matched to its cluster and poorly matched to neighboring clusters. Heat budget analysis of the ocean mixed-layer temperature tendency was used to quantify the contributions of different processes to the development of the three types of El Niño[12].

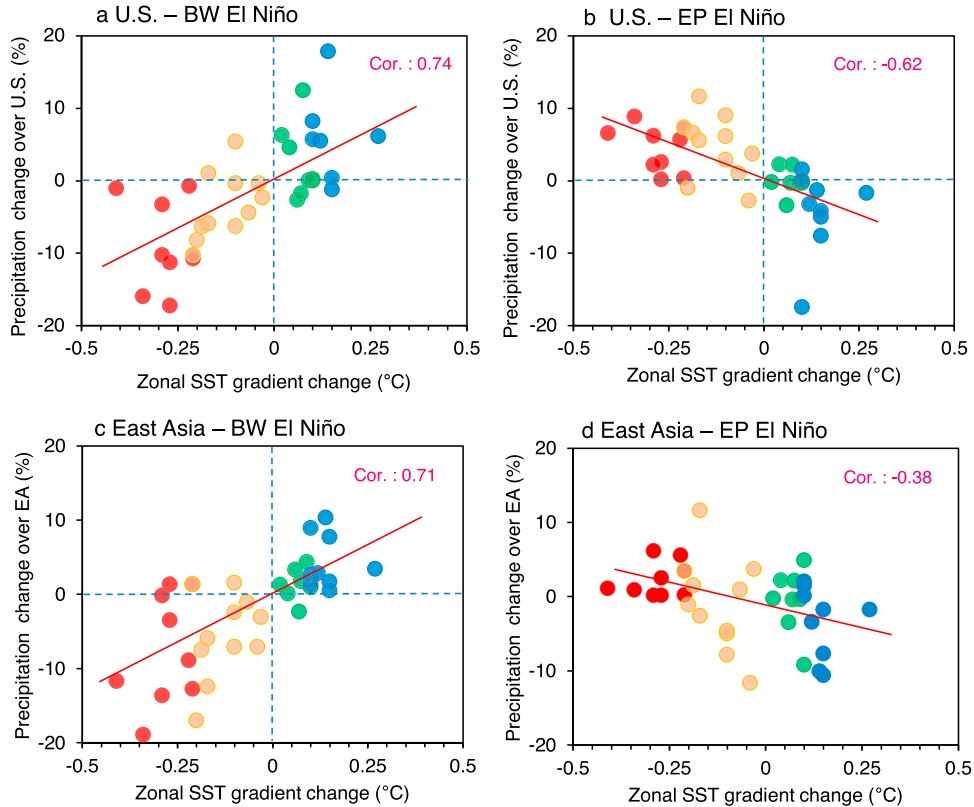

**Fig. 5 Future precipitation changes associated with zonal SST gradient change. a**, **b** Scatter plots between the changes in zonal sea surface temperature (SST) gradients over the equatorial Pacific (*x*-axis, unit: °C) and boreal winter (October to February) precipitation changes (*y*-axis, unit: %) over the central and eastern America (20°N–35°N, 110°W–75°W), induced by Basin-Wide (BW) and Eastern Pacific (EP) types of El Niño events. The zonal SST gradient is defined as the SST difference between the western Pacific (5°S–5°N, 150°E–175°W) and eastern Pacific (5°S–5°N, 115°W–145°W). Eighteen climate models with two ensemble members were used. The change indicates the difference between historical (1943–1992) and Representative Concentration Pathway (RCP) 4.5 (or RCP 8.5) (2038–2087) scenarios. **c**, **d** Same as **a**, **b**, but for East Asia (EA, 25°N–35°N, 110°E–125°E). Red (blue) dots represent the models with El Niño-like (La Niña-like) mean SST change. The linear regression line is shown as the solid red line in each panel, with correlation coefficient. Red (orange) dots represent the models experiencing strong (weak) El Niño-like mean SST changes, while blue (green) dots indicate strong (weak) La Niña-like mean SST changes in the future.

**Historical and future scenario runs of CMIP5 models**. The historical, RCP4.5, and RCP8.5 simulations from the Coupled Model Intercomparison Project phase 5 (CMIP5) were used to examine the dependence of the future changes in El Niño-induced precipitation anomalies on the changes in the zonal SST gradient in the mid-latitude land region. To quantify the performance of the CMIP5 models on the three distinct El Niño events (the BW, EP, and CP types), we used the pattern correlation coefficient (PCC) of the temporal evolution of SST anomalies (Fig. 2) between the observed and historical model simulations. We selected 18 models with relatively high PCCs (>0.45) that can capture the observed horizontal patterns of the three distinct El Niño events in their historical runs during the El Niño peak time (CNRM-CM5, CESM-WACCM, CCSM4, CanESM3, CESM-CAM5, GFDL-CM3, CESM-BCG, IPSL-CM5B-LR, NorESM1-M, BCC-CSM1–1, GFDL-ESM2M, ACCESS1–3, IPSL-CM5A-LR, MPI-ESM-MR, and MRI-CGCM3, IPSL-CM5A-MR, MPI-ESM-LR, GFDL-ESM2G). Note that all models used in this study produce reasonable El Niño simulations for the boreal winter season[25]. We conducted a K-cluster analysis on the historical, RCP4.5, and RCP8.5 simulations using the 18 models to investigate the various types of El Niño events, and estimated the precipitation anomalies worldwide during the peak time of El Niño events for all simulations.

## Data availability
All observed data used in this study are publicly available (https://psl.noaa.gov/data/gridded/data.20thC_ReanV3.html; https://psl.noaa.gov/data/gridded/data.noaa.ersst.v5.html). The new data can be downloaded here: https://figshare.com/articles/dataset/Data_ElNINO_driven_precipitation_anomalies_from_CMIP5/13351997.

## Code availability
The codes used in this study can be download here: https://figshare.com/articles/software/code-ENSO-induced-precipitation-change/13273271.

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

## Acknowledgements
This work was supported by the National Science Foundation of China (Grant No. 42088101), National Research Foundation of Korea (NRF; NRF-2018R1A5A1024958 and NRF-2020R1C1C1006569), and Nanjing University of Information Science and Technology initial research foundation (1441012001019). This is Publication No. 11200 of SOEST, Publication No. 1490 of the IPRC, and Publication No. 337 of Earth System Modeling Center.

## Author contributions
Y.-M. Yang, S.-I. An, and B. Wang conceived the idea. Y.-M. Yang performed model experiments and analyses. S.-I. An, Y.-M. Yang, B. Wang, J. H. Park, and X. Luo wrote the manuscript. All authors provided critical feedback and helped shape the research, analysis, and manuscript.

## Competing interests
The authors declare no competing interests.
