## [Peer Review File · Nature Communications]

REVIEWER COMMENTS

Reviewer #1 (Remarks to the Author):

The authors sorted global climate models into two groups based on the difference of mean-state sea surface temperature (SST) pattern changes in the tropical Pacific under greenhouse gas forcing. One group exhibits an El Niño-like mean-state change, in which SST warms more in the east. The other group exhibits a La Niña-like mean-state change, in which SST warms more in the west. Then, they investigated the difference of the El Niño Southern Oscillation (ENSO) changes and the associated teleconnection signals between the two groups. They found that, during Basin Wide (BW) El Niño events and East Pacific (EP) El Niño events, there seems to exist a constraint between the zonal SST gradient change and precipitation.

I think their results are overall novel and reasonable, and I would be happy to see this work on Nature Communications after some minor revisions, particularly regarding presentations. Specific comments are as follows.

1. Some paragraphs are too long, so they are hard to read. I would suggest splitting the paragraphs. For example, the paragraphs starting from Lines 68 and 113 take almost a full page each, and I think they are long enough to discourage the potential readers.
2. Fig. 1, 2, 3 and 4: I suggest putting labels to explain what the rows and columns mean. For example, in Fig. 1, I would put labels to clarify that the upper panels show SST changes and the lower panels show wind changes.
3. Fig. 1: What are the numbers in the brackets?
4. The upper panels of Fig. 1b: Why are the colors of MRI-CGCM3 and MPI-ESM-MR reversed between the two panels?
5. Fig. 4, caption:
(a) EP and (b) BW
=> (a) BW and (b) EP
6. Fig. 4, caption: Please explain what the top panels are. I guess they show observations. If so: The GPCP data (1900-2015) and NCEP 20C data are used for observed precipitation and circulation pattern.
=> The GPCP data (1900-2015) and NCEP 20C data are used for observed precipitation and circulation pattern (top).
7. Fig. 5, caption: In the last sentence, it is explained that "the linear regression line is shown by the dotted line with the significance indicated in each panel", but I couldn't find the information of significance.

Reviewer #2 (Remarks to the Author):

Review of Future changes in El Niño-induced midlatitude precipitation projected by mean SST change constraints by Yang et al.

In this study, the authors analyzed 20th century historical and 21st century future simulations under both RCP4.5 and RCP8.5 emissions scenario from the CMIP5 archive to quantify the changes in the 21st century (relative to the 20th century) in El Niño-induced tropical and midlatitude precipitation under two different mean SST changes: one with a negative and one with a positive

change in the zonal SST gradient in the equatorial Pacific. They first showed that the changes in the frequency and intensity of the El Niño events (categorized into three types) depend on the zonal SST gradient change (Fig. 3), and then showed that the precipitation anomaly composites associated with those El Niño events may differ over some regions such as North America and East Asia between the models with a negative SST gradient change and models with a positive SST gradient change (Figs. 4-5). While I found some of the results (e.g., Figs. 3 and 5) are interesting, the regional precipitation differences are relatively small and it is unclear whether they are significant or not statistically. The number of data points shown in Figs. 3 and 5 is quite small. As a result, I felt that the authors did not provide sufficient evidence to support their main conclusions.

In general, I found the manuscript needs major improvements. For example, the figure captions need major improvements in order for the reader to be able to understand their results. The Methods section needs to include more information regarding how they defined the anomalies and derived the El Niño composites. It is unclear whether and how they have separated the long-term changes (from the global warming) in the precipitation, SST and other fields from the El Niño-induced changes. The use of a 95-year period in the 20th and 21st century for computing the change is also unusual (and used without any justification) as it does not really represent a change from the present to the end of the 21st century. I understand they may need a long period to increase the number of El Niño events, but that would be better done by including more model ensemble runs (and it is unclear how many model ensemble runs were included for each of the 8 selected models). Also, there are no analyses done to reveal how many of the CMIP5 models show insignificant SST gradient changes, a positive, or a negative gradient change. No efforts were made to show any of the changes are statistically significant and not due to random noise. Labeling on the figures also needs improvements. Given these significant issues, I felt that the manuscript is not of high quality.

Specific comments:

1. Abstract: it is unclear how many of the climate models project a La Niña-like, El Niño-like, or neutral mean SST change by the end of the 21st century, and under what emissions scenarios. Without such basic background information, all the discussions in the Abstract, including the change numbers, are not very helpful.
2. Lines 36-37: This sentence discusses the major impact of ENSO, yet the cited refs. 1-6 appear to be on ENSO's properties (perhaps except ref. 5-6 on the impact on tropical cyclones), not its impact on precipitation (the focus of this study) and other fields over land and other regions. There are many papers on ENSO's impact on precipitation, temperature and other climatic fields; those papers should be cited here. This raises the question regarding the accuracy of their citations in the paper.
3. Line 45: While the El Niño diversity may be relevant to the impact of future El Niño, ENSO's future occurrence frequency and intensity are likely more important for its future impact on midlatitude precipitation. The discussion in this paragraph seems to focus only on a secondary issue but have missed the most important point. Please note that many papers have discussed how ENSO (i.e., its frequency and intensity) may change in the 21st century; their findings are highly relevant to the discussion here.
4. Line 63, "Using climate model analysis"  "Through analyses of climate model simulations". Need to discuss how you would separate the impacts on precipitation from the mean warming and from El Niño in a future warmer climate. If this separation is incorrect, your results and conclusions could be wrong and misleading. This is a crucial point needs to be discussed in detail before the reader can trust your results.
5. Fig. 1: It is very unusual to use the difference between 2006-2100 and 1911-2005, which can't

be interpreted as future changes relative to present climate. Most people (including IPCC reports) use the difference between a present (e.g., 1970-1999) and a future (2070-2099) period to quantify the future changes. It is unclear which panels are for SST and for surface U wind changes. There are four rows in the figure, what do your "upper panels" and "lower panels" refer to? Label your x and y-axis in this and all other figures, and label all the panels with a, b, c, d, etc. The figure caption is inaccurate and hard to follow. For example, I assume the top four panels are from models with a negative mean-state zonal SST gradient change and the lower four panels are from models with a positive mean-state zonal SST gradient change. This is very different from what is said in the caption. And your "mean-state zonal gradient SST change" is hard to understand. Please define it in the caption, and define the numbers in parentheses and avoid showing the same numbers twice (e.g., in the left top two panels and lower two panels)! I don't think this figure is very interesting.

6. Line 88-90: The SST and surface wind changes are simultaneous response of the coupled system to the future GHG forcing. They are physically consistent changes, but you can't say one is contributing to or causing the other! I'm not sure the purpose of showing Fig. 1.

7. Fig. 2: It would be nice if the observations are included in this figure, so the reader can access if the model-simulated evolutions resemble observations (or reanalysis data). You should reduce the blank space between the rows and define your x and y axis. Label the three rows with HIST, RCP4.5 and RCP8.5. I assume all the El Nino events from the four models in each group were used in the composite averaging, so you may do not need that many years (e.g., a 50 year period for 1950-1999 and 2050-2099 might be sufficient?).

8. Lines 98-138: The reader is more interested in whether the model-simulated SST evolution is realistic in comparison with observations, not the model-simulated features that may be model dependent and unrepresentative of the real world. So, please discuss the features in the context of observations.

9. Figs. 1-2 have very little to do with the topic of this study: Future El Nino-induced precipitation changes.

10. Need a better caption for Fig.3, e.g., "Scatter plots between the changes in zonal mean SST gradients over the equatorial Pacific (x axis) and (a) El Nino frequency of the (left) BW and CP and (right) EP type or (b) El Nino intensity of the (left) BW and CP and (right) EP type. ..." Fig. 3 is interesting, but it would be more convincing if there are more data points included, for example including more models or more simulations (e.g., use more than one ensemble simulations and/or include CMIP6). Because of your use of 95 years, the differences between the RCP4.5 and RCP8.5 are small as they are similar before ~2060. You need to focus on the last 20-50 years of the 21st century see their differences.

11. Line 168: "pattern correlation coefficient"? There are no spatial patterns shown in Fig. 3! Did you mean Pearson's correlation coefficient? Why don't just say "correlation coefficient (r)"?

12. Line 170, "as observed over the 20th century" needs to cite refs to support such statement here and other places.

13. Fig. 4; It is unclear which panel is from GPCP and NCEP 20C data. I don't think there is anything call NCEP 20C. It may be NOAA 20C reanalysis? Also, its v2 and v3 are very different. Include the version number in the caption. It seems to me that you used arbitrary time periods. Try to use the same historical period (1911-2005) for your GPCP and NOAA 20CR data. State the units of the variables shown, and move the color bar closer to the panels in (b)! The observational data description at lines 304-315 do not match what stated in the caption: e.g., does the GPCP dataset cover 1900-2015 or 1901-2014? What is the NCEP 20C data? Suggest to use NOAA 20CRv3 data only, not the combination of NCEP/DOE reanalysis.

14, Fig. 4 and other figures: I could not find any description in Methods or other parts of the paper regarding how the precipitation and other anomalies were defined and how the El Nino composites were derived. Given the strong long-term change signal in the model data, it seems to me that special care is needed in separating the impact from global warming and the real impact from the El Nino in deriving the El Nino composites. For example, the mean SST changes (including its zonal gradient changes) may induce precipitation changes in the tropical Pacific and land areas, but those changes are not part of the El Nino-induced changes. Thus, one needs to separate that impact first before making the El Nino composite. Otherwise, the El Nino composites may be contaminated by the different background changes that is not really part of the El Nino-induced changes. It is unclear whether this is considered and done in the present study.

15. Fig. 4: It is unclear whether the precipitation and wind changes are statistically significant or just due to sampling noise. Same applies to other figures as well.

16. Fig. 5: Again, more data points would be helpful, e.g., from more ensemble runs or from more models.

17. Lines 284-285: Fig. 4 shows some regional differences in the color shading between the left and right panels, but one needs to look very closely to find those differences, and it is unclear whether those regional differences are statistically significant. The relationship shown in Fig. 5 is also not very strong given the relatively small number of data points. Thus, I don't think the authors presented sufficient evidence to make this statement.

18. Lines 286-291: It is unclear that among all the CMIP5 models, how many of them show a positive and how many of them show a negative zonal SST gradient change, and how many of them show insignificant gradient changes? The authors need to find out this answer in order to make these statements really relevant. For example, if 90% of the CMIP5 (or CMIP6) models show a positive gradient change and only 10% show a negative gradient change, then the positive gradient results are most relevant. On the other hand, if most of the CMIP5 models show insignificant changes in the SST gradient, then the findings here are not really relevant for most models.

18. Line 303, Methods: lack of description regarding how the anomalies and El Nino composites for SST, precipitation, winds and other fields were derived. How did you remove the long-term warming signal from the El Nino signal for all the variables? From how many models you examined and then selected the eight models? Did you use one single run from each of the models? Can you use all the available ensemble runs from the selected models to increase the data points in Figs. 3 and 5?

Response to Reviewer #1 comments:

The authors sorted global climate models into two groups based on the difference of mean-state sea surface temperature (SST) pattern changes in the tropical Pacific under greenhouse gas forcing. One group exhibits an El Niño-like mean-state change, in which SST warms more in the east. The other group exhibits a La Niña-like mean-state change, in which SST warms more in the west. Then, they investigated the difference of the El Niño Southern Oscillation (ENSO) changes and the associated teleconnection signals between the two groups. They found that, during Basin Wide (BW) El Niño events and East Pacific (EP) El Niño events, there seems to exist a constraint between the zonal SST gradient change and precipitation.

I think their results are overall novel and reasonable, and I would be happy to see this work on Nature Communications after some minor revisions, particularly regarding presentations. Specific comments are as follows.

Re: Thank you for your valuable comments and suggestions. In the revised manuscript, 1) we increase data numbers to obtain more significant results, and 2) we modified all figures as the reviewer suggested. We believe that the revised manuscript provides better figures and explanations by addressing your constructive comments.

1. Some paragraphs are too long, so they are hard to read. I would suggest splitting the paragraphs. For example, the paragraphs starting from Lines 68 and 113 take almost a full page each, and I think they are long enough to discourage the potential readers.

Re: Thank you for your valuable comment. We have split long paragraphs and make each paragraph having a coherent theme. Also, a native English speaker edited the revised manuscript.

2. Fig. 1, 2, 3 and 4: I suggest putting labels to explain what the rows and columns mean. For example, in Fig. 1, I would put labels to clarify that the upper panels show SST changes and the lower panels show wind changes.

Re: we revised the figure as the reviewer mentioned. Figure R1 shows a revised figure showing El Niño evolution, where labels are added on the rows and columns. Note that figure

1 in the original manuscript was removed.

Figure R1. Longitude-time diagram of the equatorial Pacific SST anomalies under (a) El Niño- and (b) La Niña-like zonal SST gradient change. Left to right columns correspond to BW-, EP-, CP-type El Niño evolution. Each panel illustrates the composite of the equatorial SSTA (in units of $^{\circ}\text{C}$) averaged between 5°S and 5°N from observation (upper panel) and model simulations (middle and lower panel). The “present” represents the historical run (1943-1992) and the “future” shows a composite of both RCP4.5 (2038-2-2087) and RCP8.5 (2038-2087) from 18 CMIP5 models with two ensemble members. The time axis starts from October of the year before the El Niño year (–1) to the February after the El Niño year (1). All data are applied to a high-pass spectral filter (13yrs) to removed global warming and long-term climate variability. The stippling denotes the regions where the signal (group

mean) is larger than the noise (one standard deviation (SD) of each member from the group mean).

3. Fig. 1: What are the numbers in the brackets?

Re: The number represents zonal SST gradient change between historical and RCP 4. (or RCP8.5) simulations. Note that figure 1 in the original manuscript was removed because it may confuse readers and is not essential.

4. The upper panels of Fig. 1b: Why are the colors of MRI-CGCM3 and MPI-ESM-MR reversed between the two panels?

Re: We removed the original Figure 1b and replaced it in Figure R1 above.

5. Fig. 4, caption:

(a) EP and (b) BW

=> (a) BW and (b) EP

Re: Corrected

6. Fig. 4, caption: Please explain what the top panels are. I guess they show observations. If so:

The GPCC data (1900-2015) and NCEP 20C data are used for observed precipitation and circulation pattern.

=> The GPCC data (1900-2015) and NCEP 20C data are used for observed precipitation and circulation pattern (top).

Re: We modified figures and the corresponding captions as the reviewer suggested.

Figure R2. Composites of boreal winter (October to February) precipitation anomalies induced by (a) BW- and (b) EP-types of El Niño derived from observations (top) and CMIP5 models (bottom four). Left (right) panels in the CMIP5 model results show composites of the model with La Nina-like (El Niño-like) mean SST change. The “present” represent the historical run (1943-1992) and the “future” shows both RCP4.5 and RCP8.5 (2038-2087) from 18 CMIP5 models with two ensemble members. All data are applied to a high-pass spectral filter (13yrs) to removed global warming and long-term climate variability. The stippling denotes the regions where the signal (group mean) is larger than the noise (the SD of each member from the group mean). The NOAA 20th Century Reanalysis v3 dataset is used for observation.

7. Fig. 5, caption: In the last sentence, it is explained that “the linear regression line is shown by the dotted line with the significance indicated in each panel”, but I couldn’t find the information of significance.

Re: It was misprinted. We modified that sentence – “the linear regression line is shown by the solid line with a correlation coefficient”.

Figure R3. Scatter plots between the changes in zonal SST gradients over the equatorial Pacific (x-axis) and boreal winter (October to February) precipitation changes over (a) the central and eastern America (20°S–35°N, 110°W–75°W), and (b) East Asia (25°S–35°N, 110°E–125°E) induced by (left) BW- and (right) EP-El Niño events. The zonal SST gradient is defined as the SSTA difference between the western Pacific (5°S–5°N, 150°E–180°E) and eastern Pacific (5°S–5°N, 210°E–240°E). Eighteen CMIP5 models with two ensemble members were utilized. The change indicates the difference between historical (1943-1992) and RCP 4.5 (or RCP 8.5) (2038-2087). Red (blue) dots represent the models with El Niño-like (La Niña-like) mean SST change. The linear regression line is shown by the solid red line in each panel, with a significant correlation coefficient.

Reviewer #2 (Remarks to the Author):

Review of Future changes in El Nino-induced midlatitude precipitation projected by mean SST change constraints by Yang et al.

In this study, the authors analyzed 20th century historical and 21st century future simulations under both RCP4.5 and RCP8.5 emissions scenario from the CMIP5 archive to quantify the changes in the 21st century (relative to the 20th century) in El Nino-induced tropical and midlatitude precipitation under two different mean SST changes: one with a negative and one with a positive change in the zonal SST gradient in the equatorial Pacific. They first showed that the changes in the frequency and intensity of the El Nino events (categorized into three types) depend on the zonal SST gradient change (Fig. 3), and then showed that the precipitation anomaly composites associated with those El Nino events may differ over some regions such as North America and East Asia between the models with a negative SST gradient change and models with a positive SST gradient change (Figs. 4-5). While I found some of the results (e.g., Figs. 3 and 5) are interesting, the regional precipitation differences are relatively small and it is unclear whether they are significant or not statistically. The number of data points shown in Figs. 3 and 5 is quite small. As a result, I felt that the authors did not provide sufficient evidence to support their main conclusions.

In general, I found the manuscript needs major improvements. For example, the figure captions need major improves in order for the reader to be able to understand their results. The Methods section needs to include more information regarding how they defined the anomalies and derived the El Nino composites. It is unclear whether and how they have separated the long-term changes (from the global warming) in the precipitation, SST and other fields from the El Nino-induced changes. The use of a 95-year period in the 20th and 21st century for computing the change is also unusual (and used without any justification) as it does not really represent a change from the present to the end of the 21st century. I understand they may need a long period to increase the number of El Nino events, but that would be better done by including more model ensemble runs (and it is unclear how many model ensemble runs were included for each of the 8 selected models). Also, there are no analyses done to

reveal how many of the CMIP5 models show insignificant SST gradient changes, a positive, or a negative gradient change. No efforts were made to show any of the changes are statistically significant and not due to random noise. Labeling on the figures also needs

improvements. Given these significant issues, I felt that the manuscript is not of high quality.

Re: Thank you for your constructive comments and suggestions. In the revised manuscript, 1) we increased the numbers of the models used in this study up to 18 to derive a convincing relationship between projected changes in BW (or EP) El Nino or regional precipitation anomalies and zonal SST gradient changes, 2) we reduce the analysis period (last 50 years) and used two ensemble numbers to collect plenty of El Nino events, 3) We explained how the global warming signals are removed from the SST and precipitation anomalies, and how the zonal SST gradient changes are defined. We also modified all figures and captions to improve their quality and readability. We implemented several figures to explain the possible mechanisms and support the conclusions. We believe that the revised manuscript is significantly improved by addressing your valuable comments.

Specific comments:

1. Abstract: it is unclear how many of the climate models project a La Nina-like, El Nino-like, or neutral mean SST change by the end of the 21st century, and under what emissions scenarios. Without such basic background information, all the discussions in the Abstract, including the change numbers, are not very helpful.

Re: We added detailed information on the present and projected zonal SST gradient changes in CMIP5 and observations. Figure R1 shows the zonal SST gradient change from the observation and CMIP5 models during present days and future scenarios. For the ‘future’, the zonal SST gradient changes from the models are defined by the differences between historical and RCP 4.5 (or RCP8.5). The observed change (that is, the difference between a recent period (1980-2005) and a past period (1940-1970)) shows moderate warming in the western Pacific compared to the eastern Pacific (a La Nina-like SST change). In corresponding model simulations (historical runs), the models with a La Nina-like SST change account for about 40% of the total models used in this study while the model with an El Nino-like SST change is about 40%, and the model with a relatively small (or neutral) SST change is about 20%. For future scenarios, the zonal SST gradient changes are also diverse. The models with a La Nina-like (El Nino-like) SST changes are about 40% (50%). These results suggest that the future change of zonal SST gradients is uncertain. Therefore, we estimate the future change of mid-latitude precipitation anomalies induced by El Nino events based on the zonal SST gradients rather than deterministic estimation by a multi-model ensemble mean technique. We included the discussions in the “Abstract” and “Main text” of the revised manuscript.

Figure R1. A scatter plot of mean SST changes in the tropical Pacific (y-axis, unit: °C) and zonal SST gradient changes (x-axis, unit: °C) in CMIP5 models during (a) present and (b) future periods. For future periods, the differences of tropical mean SST between the mean SST in the tropics (20°S–20°N) of the RCP 4.5/8.5 (2038–2087) scenario and that of the historical simulation (1943–1992) are used. The zonal SST gradient is measured by western Pacific SST (5°S–5°N, 155°E–175°W) minus eastern Pacific SST (5°S–5°N, 145°W–115°W) where changes between RCP4.5/8.5 and historical scenarios are used. Red (or orange) and blue (or green) dots indicate models experiencing El Nino (La Nina)-like mean SST change in the future. For present periods, the differences between the mean SST (or zonal SST gradient) of 1940–1970 years and that of 1980–2005 years are used. The observations are marked by a gray line. The Extended Reconstructed Sea Surface Temperature (ERSST) version 5 dataset is used for observation.

2. Lines 36-37: This sentence discusses the major impact of ENSO, yet the cited refs. 1-6 appear to be on ENSO's properties (perhaps except ref. 5-6 on the impact on tropical cyclones), not its impact on precipitation (the focus of this study) and other fields over land and other regions. There are many papers on ENSO's impact on precipitation, temperature, and other climatic fields; those papers should be cited here. This raises the question regarding the accuracy of their citations in the paper.

Re: Thank you for your comment. We have added some papers that discuss ENSO-induced tropical and midlatitude precipitation changes under global warming (Endris et al. 2019; Yan et al. 2020; Perry et al. 2017, Wang et al. 2020; Fasullo et al. 2018; Li and Ting 2015; Power et al. 2013; Xu et al. 2017). In the revised version, we have added the following description: “Previous studies discussed future changes of precipitation anomalies induced by projected ENSO in East Africa, Maritime continent, South and East Asia, North America, and major monsoon regions by enhanced ENSO-teleconnection under greenhouse gases. However, a few studies (Perry et al. 2017, Fasullo et al. 2018, Li and Ting 2015) suggested that the signal for consistent strengthening is relatively weak across the models although the multi-model ensemble mean shows robust increases in ENSO-induced precipitation. A recent study (Wang

et al. 2020) found that in CMIP6 models, an El Niño-like eastern Pacific warming reduces North American monsoon rainfall by the equatorward shift of the inter-tropical convergence zone.”

3. Line 45: While the El Nino diversity may be relevant to the impact of future El Nino, ENSO's future occurrence frequency and intensity are likely more important for its future impact on midlatitude precipitation. The discussion in this paragraph seems to focus only on a secondary issue but have missed the most important point. Please note that many papers have discussed how ENSO (i.e., its frequency and intensity) may change in the 21st century; their findings are highly relevant to the discussion here.

Re: Following your suggestion, we have added the discussion related to the ENSO changes in intensity and frequency under greenhouse warming as follows: “Previous studies reported that the intensity and frequency of ENSO may increase under global warming, but the inter-model spreads in the projected future change of ENSO are relatively large (Cai et al. 2014; Yeh et al. 2009; Cai et al. 2012; Yeh et al. 2011; Yang et al. 2018; Cai et al. 2018; Yan et al. 2020). A few studies show an increased frequency of CP-El Niño (Yeh et al. 2009), but other studies show a more frequent occurrence of EP El Nino events and stronger SST variability in the eastern Pacific (Cai et al. 2018) under greenhouse warming. Wang et al. (2019) recently suggested that extreme El Niño, which corresponds to BW-type El Niño in the present study, may occur more frequently in the future under the La Niña-like mean SST change, but not under El Niño-like mean SST change. This study motivated us to consider zonal SST gradient change for understanding future ENSO effect on climate”.

4. Line 63, “Using climate model analysis”  “Through analyses of climate model simulations”. Need to discuss how you would separate the impacts on precipitation from the mean warming and from El Nino in a future warmer climate. If this separation is incorrect, your results and conclusions could be wrong and misleading. This is a crucial point needs to be discussed in detail before the reader can trust your results.

Re: Our analysis primarily focuses on interannual variability and ENSO effects. To removed global warming trends and the impact of long-term climate variability, we applied a high-pass spectral filter (13-year) to the CMIP5 dataset (Power et al. 2013). The first and last 13 years from the total period are discarded to remove the edge effects caused by the spectral filter. Therefore, only 50yrs of historical (1943-1992) and RCP scenarios (2038-2087) are used for

the analysis and comparison. Significances of differences between the two simulations were tested by Student's T -test.

5. Fig. 1: It is very unusual to use the difference between 2006-2100 and 1911-2005, which can't be interpreted as future changes relative to present climate. Most people (including IPCC reports) use the difference between a present (e.g., 1970-1999) and a future (2070-2099) period to quantify the future changes. **It is unclear which panels are for SST and for surface U wind changes.** There are four rows in the figure, what do your "upper panels" and "lower panels" refer to? Label your x and y-axis in this and all other figures, and label all the panels with a, b, c, d, etc. **The figure caption is inaccurate and hard to follow.** For example, I assume the top four panels are from models with a negative mean-state zonal SST gradient change and the lower four panels are from models with a positive mean-state zonal SST gradient change. This is very different from what is said in the caption. And your "mean-state zonal gradient SST change" is hard to understand. Please define it in the caption, and define the numbers in parentheses and avoid showing the same numbers twice (e.g., in the left top two panels and lower two panels)! I don't think this figure is very interesting.

Re: We agree with your comment about the period of the data. To quantify the difference between historical and RCP scenarios runs objectively, we used a longer 50-year period to compare the historical (e.g., 1943-1992) and future (2038-2087) simulations. Use of the 50-year period is increase the sample size for ENSO events. To increase the sample size of El Nino events, we also used two ensemble members of the model runs.

We also realized that figure 1 is not very informative and helpful to show 'the diversity of zonal SST gradient change from CMIP5 models. Therefore, we removed figure 1 and changed to Figure R1 that shows the tropical mean SST changes and zonal SST gradient changes between the historical and future climate.

6. Line 88-90: The SST and surface wind changes are simultaneous response of the coupled system to the future GHG forcing. They are physically consistent changes, but you can't say one is contributing to or causing the other! I'm not sure the purpose of showing Fig. 1.

Re: Thank you. We have deleted those statements and removed the zonal mean wind change in figure 1.

7. Fig. 2: It would be nice if the observations are included in this figure, so the reader can

access if the model-simulated evolutions resemble observations (or reanalysis data). You should reduce the blank space between the rows and define your x and y axis. Label the three rows with HIST, RCP4.5 and RCP8.5. I assume all the El Nino events from the four models in each group were used in the composite averaging, so you may do not need that many years (e.g., a 50 year period for 1950-1999 and 2050-2099 might be sufficient?).

Re: Following your suggestion, we have included the observed El Nino diversity. We also modified the figure 2 (see Figure R2 below). First, we used 18 models. Second, we reduced periods of the dataset from 95 yrs to 50yrs as you suggested. Furthermore, we used two ensemble members for each model to increase the numbers of different types of El Nino events. Therefore, figure 2 is now based on the composite El Nino events from 18 models with two ensemble numbers. Third, we rearranged the location of each panel and added labels showing “OBS, Present, and Future”.

Figure R2. Longitude-time diagram of the equatorial Pacific SST anomalies under (a) El Niño- and (b) La Niña-like zonal SST gradient change. Left to right columns correspond to BW-, EP-, CP-type El Niño evolution. Each panel illustrates the composite of the equatorial SSTA (in units of °C) averaged between 5°S and 5°N from observation (upper panel) and model simulations (middle and lower panels). The “present” represents the historical run (1943-1992) and the “future” shows a composite of both RCP4.5 (2038-2087) and RCP8.5 (2038-2087) from 18 CMIP5 models with two ensemble members. The time axis starts from October of the year before the El Niño year (−1) to the February after the El Niño year (1). All data are applied to a high-pass spectral filter (13yrs) to removed global warming and long-term climate variability. The stippling denotes the regions where the signal (group mean) is larger than the noise (one standard deviation (SD) of each member from the group mean).

8. Lines 98-138: The reader is more interested in whether the model-simulated SST evolution is realistic in comparison with observations, not the model-simulated features that may be model dependent and unrepresentative of the real world. So, please discuss the features in the context of observations.

Re: We added the description for the observed El Niño evolution and compared the model simulation with the observation – “Figure 2a and 2b shows the evolutionary composites of three distinct El Niño events in the observation and the models with an El Niño-like SST change (EN-Models). The observation shows that, in the EP El Niño events, cold anomalies occur during the previous winter and spring. They drastically change to warm anomalies from the far eastern Pacific during early summer and then propagate westward and reach the maximum in December around 130°W. These changes deepen the thermocline depth in the eastern Pacific, inducing rapid warming from the eastern to central Pacific (Supplementary Figure 3a). Compared to the observation, the historical simulations from the models capture the observed evolution of equatorial SST: cold anomalies in the spring and warm anomalies from summer to winter season, westward propagating of warm SST. However, the maximum of the warm anomalies is relatively weak and shifts westward. When anthropogenic forcings were exerted in the models, EP-El Niño events tend to intensify. The stronger anomalous westerlies in the eastern Pacific (Supplementary Figure 2a) may induce a deeper thermocline depth (Supplementary Figure 3a).

In observation, BW El Niño events start in the western Pacific during the previous winter and propagate eastward with a rapid basin-wide extension. BW El Niño shows a very strong maximum amplitude in the eastern Pacific. The models seem to reproduce the observed pattern of SST anomalies reasonably: the initiation of warm anomalies in the western Pacific,

eastward propagation, and rapidly increased warm SST anomalies. However, the models slightly overestimate the maximum of the SST anomalies and simulate a long-lived peak in the winter season. The westerly anomalies occur in the initial phase (e.g., westerly wind burst events) and both SST warming and a deep thermocline are generated, which move into the central Pacific. The BW El Niño may be developed by both zonal advective feedback in the central Pacific and thermocline feedback in the eastern Pacific. To examine the dominant dynamic processes of three El Niño types, we have conducted an ocean mixed-layer heat budget analysis (see Method in the revised manuscript). It is shown that the zonal advective and thermocline feedback is much stronger in the BW El Niño event (Supplementary Table 1). Under the global warming scenario, the warm SST anomalies are reduced significantly and become meridionally narrow in the equatorial eastern Pacific. The El Niño-like mean SST change induces weakened anomalous westerlies in the western and central Pacific by weakening-associated convective anomalies (Supplementary Figure 2a). These changes may reduce zonal advective feedback in the western Pacific and thermocline feedback in the eastern Pacific, resulting in a decrease in the intensity of the BW type of El Niño.

For CP El Niño, the observation shows similar temporal evolution to that of BW El Niño, but their magnitudes are relatively weak. The models capture the initiation of warm SST anomalies in the western Pacific but fail to simulate eastward propagation; the model shows both eastward from the western Pacific and westward propagation in the eastern Pacific. The changes in SST in CP El Niño caused by global warming are also similar to those of BW El Niño. The magnitude of CP El Niño tends to reduce with increasing anthropogenic forcings, indicating that the change of CP El Niño may be affected by the changes in the zonal SST gradient.

Figure 2c shows a composite of El Niño events from the model with La Niña-like mean SST change. The temporal evolutions of the BW and CP El Niño events from the historical runs resemble those of the EN-Models, but their magnitudes during the mature phase are relatively weak. With moderate anthropogenic forcings, the warm SST anomalies of the BW (or CP) types of El Niño are intensified and broadened outward from the equator (Supplementary Figure 1b). The cold anomalies in the western Pacific and the eastern Indian Ocean are slightly reduced. Stronger anomalous westerlies in the western Pacific may contribute to more intensive BW (or CP) El Niño. These results suggest that positive change in the zonal SST gradient may lead to strong BW (or CP) El Niño events via enhanced zonal advective feedback.

The evolutionary structure of EP El Niño events is quite similar to that of the EN-Models. In the global warming simulation, the magnitudes of SST anomalies are weakened and westward extension of the warm anomalies is reduced. The reduced mean westerly surface wind in the eastern Pacific may contribute to weak EP El Niño by reducing the thermocline and enhancing upwelling feedback.”

9. Figs. 1-2 have very little to do with the topic of this study: Future El Nino-induced precipitation changes.

Re: We reduce the explanation of figure 1 and figure 2 and more concentrated on the discussion for figure 3-5 in the revised manuscript. Our hypothesis is that future changes of precipitation in the mid-latitudes may be dependent on the intensity and frequency changes of BW (or EP) El Nino events, which may be determined by mean SST change. First, we showed that mean SST changes may be critical for the change of BW (or EP) El Nino events using figure R2 and figure R3. Second, we discuss possible precipitation changes by mean SST change through the change of BW (or EP) El Nino events using figure R4, figure R5 and figure R6.

10. Need a better caption for Fig.3, e.g., “Scatter plots between the changes in zonal mean SST gradients over the equatorial Pacific (x axis) and (a) El Nino frequency of the (left) BW and CP and (right) EP type or (b) El Nino intensity of the (left) BW and CP and (right) EP type. ...” Fig. 3 is interesting, but it would be more convincing if there are more data points included, for example including more models or more simulations (e.g., use more than one ensemble simulations and/or include CMIP6). Because of your use of 95 years, the differences between the RCP4.5 and RCP8.5 are small as they are similar before ~2060. You need to focus on the last 20-50 years of the 21st century see their differences.

Re: First, we extended the number of the models to 18 with two ensemble members (36 points from RCP4.5-Historical and RCP8.5-Historical) to derive a more convincing relationship between zonal SST gradient change and modulation of BW/CP/EP-El Niño intensity (or frequency), as the reviewer suggested. Second, we also reduced the period of the historical (1933-1992) and RCP 4.5 (or RCP8.5) (2038-2087). Third, we modified the caption of figure 3 as the reviewer suggested. As shown in Figure R3, when the zonal SST gradient increases under anthropogenic forcing, both the intensity and frequency of BW types of El Niño events increase significantly with a high correlation coefficient of 0.76 and 0.70,

respectively. Similarly, the intensity (or frequency) of CP type El Niño increase in proportional to the zonal SST gradient. This implies that if anthropogenic forcings induce La Nina-like warming in the equatorial Pacific, as observed over the 20th century (Wang et al. 2019, Kohayama et al. 2017), BW (and/or CP) El Niño events will occur more frequently. On the other hand, if the magnitude of the zonal SST gradient decreases, the frequency and intensity of the EP type of El Niño will increase with a correlation coefficient of -0.74 and -0.61, respectively.

Figure R3. Scatter plot between (x-axis) zonal SST gradient changes over equatorial Pacific and (y-axis) (a) frequency / (b) intensity change of the (left) BW, (middle) EP, and (right) CP types El Niño, respectively. The zonal SST gradient is defined as the SST difference between the western Pacific (5°S–5°N, 155°E–175°E) and eastern Pacific (5°S–5°N, 145°E–115°E). The intensity is measured by boreal winter (October to February) SST anomalies averaged over 5°S–5°N and 80°W–180°W. The eighteen CMIP5 models were used. The frequency and intensity changes are differences between historical (1943-1992) and RCP 4.5 (or RCP 8.5) (2038-2087). Red (blue) dots represent the models with El Niño-like (La Niña-like) mean SST change. The linear regression line is shown by the solid red line in each panel, with a correlation coefficient.

11. Line 168: “pattern correlation coefficient”? There are no spatial patterns shown in Fig. 3!

Did you mean Pearson's correlation coefficient? Why don't just say "correlation coefficient (r)"?

Re: It was misprinted. we modified the expression as to "correlation coefficient".

12. Line 170, "as observed over the 20th century" needs to cite refs to support such statement here and other places.

Re: We added references: (Wang et al. 2019, Kohayama et al. 2017, Lian et al, 2018)

13. Fig. 4; It is unclear which panel is from GPCC and NCEP 20C data. I don't think there is anything call NCEP 20C. It may be NOAA 20C reanalysis? Also, its v2 and v3 are very different. Include the version number in the caption. It seems to me that you used arbitrary time periods. Try to use the same historical period (1911-2005) for your GPCC and NOAA 20CR data. State the units of the variables shown, and move the color bar closer to the panels in (b)! The observational data description at lines 304-315 do not match what stated in the caption: e.g., does the GPCC dataset cover 1900-2015 or 1901-2014? What is the NCEP 20C data? Suggest to use NOAA 20CRv3 data only, not the combination of NCEP/DOE reanalysis.

Re: We used NOAA 20th Century Reanalysis (v3) data only with the same historical period (1943-1992) in the revised manuscript. We also modified the related captions and figures using new precipitation data, as shown in Figure R4, and added observational data description in the revised manuscript—"Figure R4a shows a composite of precipitation anomalies from LN-Models during BW El Niño events. Observations show strong, dry anomalies in Australia, the Amazon, and South Africa and wet anomalies in the central and eastern U.S., East Asia, and East Africa. The climate models can capture observed historical precipitation anomalies over both tropical and mid-latitude land areas. In tropical oceans, strong, wet anomalies are seen in the central and eastern Pacific while dry anomalies are seen in the western Pacific, eastern Indian Ocean, and the tropical Atlantic Ocean. In the tropics, the horizontal patterns of precipitation anomalies are similar to those of corresponding SST anomalies (e.g., Supplementary Figure 1a). Regarding tropical land, wet anomalies occur in West Africa and dry anomalies occur in the Amazon. The warm SST anomalies associated with BW El Niño generate rising motions, with strong upper-level divergent flows that are connected to upper-level convergences in the western Pacific and eastern Indian ocean, as well as the tropical Atlantic Ocean, where sinking motions are dominant (Supplementary Figure 4a). Those descending motions may contribute to suppressing convection and thus less

precipitation. The descending motions in the western Pacific produce surface easterly anomalies over the Indian Ocean, which enhances the transport of moisture to Central Africa and may lead to wet precipitation anomalies. The anomalous descending branch of Walker circulation over the equatorial western Pacific gradually builds up with the aid of the colder SST anomalies and low-level divergence, reinforcing warming in the central and eastern Pacific. In mid-latitudes, there are wet anomalies in East Asia, North America, and Eastern Europe. The descending motions in the western Pacific and the tropical Atlantic Ocean generate anticyclonic wind anomalies, and induce northward transport of moisture from the tropics to mid-latitude regions, which may contribute to more precipitation over East Asia and the eastern part of North America.”

Figure R4. Composites of boreal winter (October to February) precipitation anomalies induced by (a) BW and (b) EP types of El Niño derived from observations (top) and CMIP5 models (middle and bottom). Left (right) panels show composites of the model with La Nina-like (El Nino-like) mean SST change. The “present” represent historical run (1943-1992) and the “future” shows both RCP4.5 and RCP8.5 (2038-2-2087) from 18 CMIP5 models with two ensemble members. All data are applied to a high-pass spectral filter (13yrs) to removed global warming and long-term climate variability. The stippling denotes the regions where the signal (group mean) is larger than the noise (the SD of each member from the group mean).

14, Fig. 4 and other figures: I could not find any description in Methods or other parts of the paper regarding how the precipitation and other anomalies were defined and how the El Nino composites were derived. Given the strong long-term change signal in the model data, it seems to me that special care is needed in separating the impact from global warming and the real impact from the El Nino in deriving the El Nino composites. For example, the mean SST changes (including its zonal gradient changes) may induce precipitation changes in the tropical Pacific and land areas, but those changes are not part of the El Nino-induced changes. Thus, one needs to separate that impact first before making the El Nino composite. Otherwise, the El Nino composites may be contaminated by the different background changes

that is not really part of the El Niño-induced changes. It is unclear whether this is considered and done in the present study.

Re: To focus on the interannual ENSO influence on the precipitation, we applied a high-pass spectral filter (13yrs) to remove the global warming trend, and long-term climate variability (Power et al. 2013). The first and last 13 years of the data sets are discarded to exclude edge effects induced by the spectral filter and only 50-year of historical (1943-1992) and RCP scenarios (2038-2087) are used for the analysis of precipitation and El Niño events. To obtain enough number of El Niño events, we use two ensemble members (100 years are used for analysis). The same methods are applied to the observational analysis.

To check whether the El Niño-induced precipitation changes are properly separated from the precipitation change by mean SST change under a warm climate, we calculated the precipitation anomalies during ‘normal (or neutral) years’ from the models (Figure R5). In the historical run, the models simulate wet anomalies in the western Pacific but dry anomalies in the eastern Pacific. However, the precipitation anomalies are very weak in the mid-latitude land regions (e.g. east Asia or the central U.S.). Under global warming, the results show that there is no significant precipitation change in the mid-latitude land regions between historical and future climate simulation, suggesting that the precipitation changes show in Figure R4 may be generated by the El Niño change projected by global warming rather than the mean SST changes.

Figure R5. Composites of boreal winter (October to February) precipitation anomalies during ‘normal’ or ‘neutral’ years derived from observations (top) and CMIP5 models (middle and bottom). The normal years is defined if absolute values of October-February SST anomalies 5°N – 5°S , 120°W – 170°W over was smaller than 0.4°C . Left (right) panels show composites of the model with La Nina-like (El Niño-like) mean SST change. The “present”

represent the historical run (1943-1992) and the “future” shows both RCP4.5 and RCP8.5 (2038-2-2087) from 18 CMIP5 models with two ensemble members. We used linearly detrended data. The stippling denotes the regions where the signal (group mean) is larger than the noise (the SD of each member from the group mean).

15. Fig. 4: It is unclear whether the precipitation and wind changes are statistically significant or just due to sampling noise. Same applies to other figures as well.

Re: We increase the number of the model to 18, and marked areas where the signal (group mean) is larger than the noise (the standard deviation of each member from the group mean) with dots.

16. Fig. 5: Again, more data points would be helpful, e.g., from more ensemble runs or from more models.

Re: We modified the figure by including more CMIP5 models (total 18-model) to derive a more significant relationship (Figure R5, below). We believe that the extension of the model number improves the reliability of the relationship between zonal SST gradient change and El Niño-induced precipitation change in the mid-latitude.

Figure R6. Scatter plots between the changes in zonal SST gradients over the equatorial

Pacific (x-axis) and boreal winter (October to February) precipitation changes over (a) the central and eastern America (20°S–35°N, 110°W–75°W), and (b) East Asia (25°S–35°N, 110°E–125°E) induced by (left) BW- and (right) EP-El Niño events. The zonal SST gradient is defined as the SSTA difference between the western Pacific (5°S–5°N, 150°E–180°E) and eastern Pacific (5°S–5°N, 210°E–240°E). Eighteen CMIP5 models with two ensemble members were utilized. The change indicates the difference between historical (1943-1992) and RCP 4.5 (or RCP 8.5) (2038-2087). Red (blue) dots represent the models with El Niño-like (La Niña-like) mean SST change. The linear regression line is shown by the solid red line in each panel, with a significant correlation coefficient.

From the projected precipitation change classified by each type of El Niño and from the zonal SST gradient change, we hypothesized that changes in precipitation in North America and East Asia during BW- and EP-El Niño events may be dependent on changes in the zonal SST gradient. We tested this hypothesis using the historical runs and future projections of precipitation with 18 CMIP5 models (Figure R6, below). As the zonal SST gradient increases, the intensity of precipitation by BW-El Niño events increases significantly in East Asia and North America with a correlation coefficient of 0.74 and 0.71, respectively. This implies that if anthropogenic forcing enhances the zonal SST gradients in the central Pacific, more precipitation anomalies will be expected over East Asia and North America by intensified and frequent BW-El Niño. On the other hand, for EP-El Niño events, precipitation in North America is reduced with the increasing zonal SST gradient with a correlation coefficient of -0.62. Notably, there are less significant changes in precipitation in East Asia with a correlation coefficient of -0.38.

17. Lines 284-285: Fig. 4 shows some regional differences in the color shading between the left and right panels, but one needs to look very closely to find those differences, and it is unclear whether those regional differences are statistically significant. The relationship shown in Fig. 5 is also not very strong given the relatively small number of data points. Thus, I don't think the authors presented sufficient evidence to make this statement.

Re: We replaced the Figure 4 with Figure R4 (above) by using more CMIP5 models. In Figure R4, we showed observation, present (historical), and future (composite of both RCP4.5 and RCP8.5). We also marked the area where the signal (group mean) is larger than the noise (the SD of each member from the group mean) with stippling.

18. Lines 286-291: *It is unclear that among all the CMIP5 models, how many of them show a*

positive and how many of them show a negative zonal SST gradient change, and how many of them show insignificant gradient changes? The authors need to find out this answer in order to make these statements really relevant. For example, if 90% of the CMIP5 (or CMIP6) models show a positive gradient change and only 10% show a negative gradient change, then the positive gradient results are most relevant. On the other hand, if most of the CMIP5 models show insignificant changes in the SST gradient, then the findings here are not really relevant for most models.

Re: Approximately 40% of CMIP5 models shows a positive zonal SST gradient change under greenhouse warming, whereas approximately 50% shows a negative zonal SST gradient change. The model with a relatively small (or neutral) zonal SST gradient change is about 10%. These results suggest that the zonal SST gradient change is uncertain, considering that observation undergoes a positive zonal SST gradient change during recent decades. So, it would be better to show the El Niño-induced precipitation change by considering zonal SST gradient change, rather than deterministic expression by a multi-model ensemble mean with all CMIP5 models.

19. Line 303, Methods: lack of description regarding how the anomalies and El Nino composites for SST, precipitation, winds and other fields were derived. How did you remove the long-term warming signal from the El Nino signal for all the variables? From how many models you examined and then selected the eight models? Did you use one single run from each of the models? Can you use all the available ensemble runs from the selected models to increase the data points in Figs. 3 and 5?

Re: We applied a high-pass spectral filter (13yrs) to remove climatological mean, global warming effect, and long-term climate variability (Power et al. 2013). The first and last 13 years of the data sets are discarded to remove edge effects induced by the spectral filter and only 50yrs of historical (1943-1992) and RCP scenarios (2038-2087) are used for precipitation and El Niño events. To obtain enough El Niño events, we use two ensemble members (total 100-year data are used for analysis). The same methods are applied to the observation data.

References

1. Endris, H.S., Lennard, C., Hewitson, B. et al. Future changes in rainfall associated with ENSO, IOD and changes in the mean state over Eastern Africa. *Clim Dyn* 52, 2029–2053

- (2019). <https://doi.org/10.1007/s00382-018-4239-7>
2. Yan, Z. et al. Eastward shift and extension of ENSO-induced tropical precipitation anomalies under global warming. *Science Advanced*, 6(2), 2020, eaax4177.
 3. Perry, S. J., McGregor, S., Sen Gupta, A. & England, M. H. Future changes to El Niño–Southern Oscillation temperature and precipitation teleconnections. *Geophys. Res. Lett.* 44, 10608–10616 (2017).
 4. Wang, B., Luo X., and Liu, J. How Robust is the Asian Precipitation–ENSO Relationship during the Industrial Warming Period (1901–2017)? *J. Clim.* 33, 2779–2792 (2020).
 5. Fasullo, J., Otto-Bliesner, B., & Stevenson, S. ENSO'S changing influence on temperature, precipitation, and wildfire in a warming climate. *Geophys. Res. Lett.* 45, 9216–9225 (2018).
 6. Li, X., and M. Ting (2015), Recent and future changes in the Asian monsoon-ENSO relationship: Natural or forced?, *Geophys. Res. Lett.*, 42, doi:10.1002/2015GL063557.
 7. Power, S., Delage, F., Chung, C., Kociuba, G. & Keay, K. Robust twenty-first-century projections of El Niño and related precipitation variability. *Nature* 502, 541–545 (2013).
 8. Xu, K., Tam, C. Y., Zhu, C., Liu, B. & Wang, W. CMIP5 projections of two types of El Niño and their related tropical precipitation in the Twenty-First Century. *J. Climate* 30, 849–864 (2017).
 9. Wang, B. et al. Historical change of El Niño properties sheds light on future changes of extreme El Niño. *Proc. Natl. Acad. Sci. U.S.A.* 116, 22512–22517 (2019).
 10. Cai, W. et al. Increasing frequency of extreme El Niño events due to greenhouse warming. *Nat. Clim. Chang.* 4, 111–116 (2014).
 11. Cai, W. et al. More extreme swings of the South Pacific convergence zone due to greenhouse warming. *Nature* 488, 365–369 (2012).
 12. Yeh, S.-W., Kirtman, B. P., Kug, J.-S., Park, W. & Latif, M. Natural variability of the central Pacific El Niño event on multi-centennial timescales. *Geophys. Res. Lett.* 38, L02704 (2011).
 13. Yeh, S.-W. et al. El Niño in a changing climate. *Nature* 461, 511–514 (2009).
 14. Yang, S., Z. Li, J.-Y. Yu, X. Hu, W. Dong, and S. He, El Niño–Southern Oscillation and its impact in the changing climate, *Nati. Sci. Rev.*, 5 (6), November 2018,
 15. Yan, Z. et al. Eastward shift and extension of ENSO-induced tropical precipitation anomalies under global warming. *Science Advanced*, 6(2), 2020, eaax4177.

16. Kohyama, T., Hartmann, D. L. & Battisti, D. S. La Niña-like mean-state response to global warming and potential oceanic roles. *J. Climate* 30, 4207–4225 (2017).
17. Lian, T., Chen, D., Ying, J., Huang, P., & Tang, Y. Tropical Pacific trends under global warming: El Niño-like or La Niña-like? *National Science Review*, 5(6), 810– 812 (2018).

REVIEWERS' COMMENTS

Reviewer #1 (Remarks to the Author):

The authors responded to my previous comments well. I would be happy to see this work on Nature Communications in its current form.

Tsubasa Kohyama

Reviewer #2 (Remarks to the Author):

I thank the authors for their efforts to adequately address most of my previous concerns (although I wish they have stated what revision was made to the manuscript in their response to each of my comments). The authors included more models and runs and re-made many of the figures. I felt the manuscript is substantially improved. However, the writing (including Abstract and figure captions) can still be improved. For example, on line 20, "half of CMIP5" does not make sense. I think it meant "half of the CMIP5 models analyzed here". Because there are three possibilities (El Nino-like, La Nina-like and neutral), mentioning only the fraction of the models showing La Nina-like response is not sufficient for the reader to figure out the fraction of the models with El Nino-like response. Better to point out both. The Abstract will need to be rewritten since it greatly exceeds the length allowed by this journal.

Other examples needing improvements:

1. Citing a large number of references together (e.g., on line 45) is a bad practice. It's better to cite individual refs for specific aspects of a statement. That is, the authors need to improve their accuracy in citing the references (by knowing the references better), an issue raised in my previous review.
2. Avoid using inaccurate words to reduce confusion. For example, in Fig. 1, the caption says "Scatter plot of mean SST change ...", which would normally mean a SST change from present to future. I think what they showed is the interannual variations, not real changes.

I hope the coauthors on this paper would spend some time to help the lead author to improve the writing and presentation of the manuscript.

Response to Reviewer #1 comments:

The authors responded to my previous comments well. I would be happy to see this work on Nature Communications in its current form.

Tsubasa Kohyama

Re: We thank Dr. Kohyama for your contribution to the peer review of this work. We appreciate it.

Reviewer #2 (Remarks to the Author):

I thank the authors for their efforts to adequately address most of my previous concerns (although I wish they have stated what revision was made to the manuscript in their response to each of my comments). The authors included more models and runs and re-made many of the figures. I felt the manuscript is substantially improved. However, the writing (including Abstract and figure captions) can still be improved. For example, on line 20, "half of CMIP5" does not make sense. I think it meant "half of the CMIP5 models analyzed here". Because there are three possibilities (El Nino-like, La Nina-like, and neutral), mentioning only the fraction of the models showing La Nina-like response is not sufficient for the reader to figure out the fraction of the models with El Nino-like response. Better to point out both. The Abstract will need to be rewritten since it greatly exceeds the length allowed by this journal.

Re: Thank you for your valuable comments and suggestions. We appreciate it. In the revised manuscript, 1) we carefully revised the abstract and the main text with more suitable and precise expressions. 2) we rewrote all captions appropriately as you requested. All authors have contributed to improving the manuscript. We believe that the revised manuscript provides better explanations by addressing your constructive comments. Also, we observed nature communications rules for abstract, main text, and figures.

Particularly, the abstract is modified as the reviewer kindly suggested– “El Niño-driven precipitation profoundly affects high population regions, although its projected future change is uncertain. We postulate that the projected mean sea surface temperature (SST) change is a critical factor for estimating future changes to the precipitation associated with El Niño events over the Northern Hemisphere. Approximately half of the climate models analyzed here exhibit “La Niña-like” mean SST changes, suggesting an uncertainty in future SST change. La Niña-like mean-state SST changes induce more intense basin-wide El Niño events with anti-cyclonic flows in the western Pacific and the tropical Atlantic Ocean, enhancing the northward transport of moisture to East Asia and North America and more precipitation in those regions. Here, we show the future precipitation induced by El Niño events is approximately 20% higher than the present precipitation over East Asia and North America if La Niña-like mean SST changes occur under anthropogenic forcing, as observed over the past century”.

Other examples needing improvements:

1. Citing a large number of references together (e.g., on line 45) is a bad practice. It's better to cite individual refs for specific aspects of a statement. That is, the authors need to improve their accuracy in citing the references (by knowing the references better), an issue raised in my previous review.

Re: Thank you for the comments again. As you suggested, we cited individual references for specific points of a statement. Below is the revised introduction.

"Considering the severe impact of the El Niño-Southern Oscillation (ENSO) on global¹⁻⁴ and regional weather^{2,5,6}, climate^{2,7}, and society^{1,7}, the improvement in the reliability of future projections of El Niño events and uncovering their underlying mechanism is of great importance to the climate science community⁸⁻¹². Many studies reported that ENSO's intensity and frequency may increase under greenhouse warming¹⁰⁻¹⁴, however, the inter-model spreads in the projected future change of ENSO is relatively large^{11,13}. A few studies show an increased frequency of central Pacific (CP) El Niño events^{9,12}, whereas other studies show a more frequent occurrence of eastern Pacific (EP) El Niño events^{10,15} and stronger SST variability in the eastern Pacific under greenhouse warming^{10,13,14}. These results suggest that there is considerable uncertainty in the prediction of future changes to El Niño diversity^{11,16,17}.

ENSO-driven tropical precipitation anomalies are projected to increase significantly under the Intergovernmental Panel on Climate Change's high-emission scenario^{19,23,24}. Previous studies discussed future changes in precipitation anomalies induced by projected ENSO in East Africa¹⁸⁻²² the Maritime continent²⁶⁻²⁸ South and East Asia^{18,23,25} North America^{18,20,24} and major monsoon regions^{18,23} by enhanced ENSO-teleconnection under greenhouse gases. However, a few studies suggested that the signal for consistent strengthening is relatively weak across the models^{20,22,23}, although the multi-model ensemble mean shows robust increases in ENSO-induced precipitation. A recent study found that in CMIP6 models, an El Niño-like eastern Pacific warming reduces North American monsoon rainfall as a result of the equatorward shift of the inter-tropical convergence zone²³. However, future changes in El Niño-induced precipitation in mid-latitude land regions have not been well addressed because there are large uncertainties in the estimation of sea surface temperature (SST) change.

Recent studies have shown that the El Niño behavior of climate models that predict La

Niña-like mean SST change in a future warmer world differs from those that predict El Niño-like mean state change, mainly because of the differences in the zonal SST gradients and upper-ocean stratification²⁹. Historically, observed warming in the western Pacific may induce more frequent and extreme El Niño events with warm anomalies over the central and eastern Pacific¹³. These results point to the importance of mean SST change for the accurate prediction of future changes in El Niño-induced precipitation anomalies. Here, we show that mean SST changes may be critical for the control of future El Niño-induced precipitation changes over the tropics and mid-latitude regions. Note that our analysis primarily focuses on interannual variability and the ENSO effect. To remove the impact of global warming and long-term climate variability (see Methods), we used a high-pass filtered climate model dataset.

2. Avoid using inaccurate words to reduce confusion. For example, in Fig. 1, the caption says "Scatter plot of mean SST change ...", which would normally mean a SST change from present to future. I think what they showed is the interannual variations, not real changes.

Re: Thank you for the constructive comments. According to your comment, all captions, including Figure 1, in the revised manuscripts are modified correctly, not to make confusion. Figure 1a shows a scatter plot between zonal SST gradient change and tropical mean SST change for observation and historical run in 18 CMIP5 models. For the zonal SST gradient change, firstly zonal SST gradient is measured by western Pacific mean SST (5°S–5°N, 155°E–175°W) minus eastern Pacific mean SST (5°S–5°N, 115°W–145°W). Then, its change is obtained by the difference between the two periods (1980–2005 minus 1940–1970, x-axis). For the tropical mean SST change, tropical mean SST (20°S–20°N) difference between two periods (1980–2005 minus 1940–1970, y-axis) are applied. Figure 1b is the same as Figure 1a but with a future period (RCP4.5/8.5) from CMIP5 models. Herein, for zonal SST gradient change, differences of zonal SST gradient between RCP4.5/8.5 (2038–2087) and historical (1943–1992) runs are applied. For tropical mean SST change, the differences of tropical mean SST between the RCP 4.5/8.5 (2038–2087) and historical (1943–1992) runs are used. Thus, these figures show differences in zonal SST gradient and tropical mean SST between two periods (not interannual variability). Regarding this, the caption of Fig. 1 is revised as below. Additionally, the relevant sentences in the manuscript are revised appropriately.

Figure 1. Two projections of mean SST gradient change. **a**, The relationship between mean SST changes in the tropical Pacific (y-axis, unit: °C) and zonal SST gradient changes (x-axis, unit: °C) in CMIP5 models during present periods. **b**, Same as a but for future periods. For present periods, the differences between the mean SST (or zonal SST gradient) of 1940–1970 years and that of 1980–2005 years are used. The zonal SST gradient is measured by western Pacific SST (5°S–5°N, 155°W–175°W) minus eastern Pacific SST (5°S–5°N, 115°W–145°W). The Extended Reconstructed Sea Surface Temperature (ERSST) version 5 dataset is used for observation. The observations are marked with a gray line. For future periods, the differences of tropical mean SST between the mean SST in the tropics (20°S–20°N) of the RCP 4.5/8.5 (2038–2087) scenario and that of the historical simulation (1943–1992) are used. The zonal SST gradient changes are defined as the differences of zonal SST gradient between RCP4.5/8.5 and historical scenarios are used. Red (or orange) and blue (or green) dots indicate models experiencing El Niño (La Niña)-like mean SST change in the future.

I hope the coauthors on this paper would spend some time to help the lead author to improve the writing and presentation of the manuscript.

Re: All authors contribute to improving the writing and accurate expression.